# A Low-Current and Multi-Channel Chemiresistor Array Sensor Device

**DOI:** 10.3390/s22072781

**Published:** 2022-04-05

**Authors:** Zaiqi Wang, Guojun Shang, Dong Dinh, Shan Yan, Jin Luo, Aimin Huang, Lefu Yang, Susan Lu, Chuan-Jian Zhong

**Affiliations:** 1Xiamen Institute of Measurement and Testing, Xiamen 361004, China; wzq@xmjly.com; 2Department of Chemistry, State University of New York at Binghamton, Binghamton, NY 13902, USA; gshang1@binghamton.edu (G.S.); syan4@binghamton.edu (S.Y.); jluo@binghamton.edu (J.L.); 3Watson Engineering School, State University of New York at Binghamton, Binghamton, NY 13902, USA; ddinh2@binghamton.edu (D.D.); slu@binghamton.edu (S.L.); 4Xiamen Smart NanoSensing Technologies, Xiamen 361021, China; aimin.huang@snsingtech.com (A.H.); lefu.yang@snsingtech.com (L.Y.)

**Keywords:** low-current multichannel electronic board, chemiresistor array, nanostructured sensing film, volatile organic compound

## Abstract

This paper describes the design of a low-current, multichannel, handheld electronic device integrated with nanostructured chemiresistor sensor arrays. A key design feature of the electronic circuit board is its low excitation current for achieving optimal performance with the arrays. The electronics can rapidly acquire the resistances for different sensors, not only spanning several orders of magnitude, but also as high as several hundreds of megaohms. The device tested is designed using a chemiresistor array with nanostructured sensing films prepared by molecularly-mediated assemblies of gold nanoparticles for detection. The low-current, wide-range, and auto-locking capabilities, along with the effective coupling with the nanostructured chemiresistor arrays, meet the desired performances of a low excitation current and low power consumption, and also address the potential instability of the sensors in a complex sensing environment. The results are promising for potential applications of the device as a portable sensor for the point-of-need monitoring of air quality and as a biosensor for point-of-care human breath screening for disease biomarkers.

## 1. Introduction

Nanostructured sensing films have been widely utilized for chemiresistor sensing in environmental and healthcare applications [1,2,3,4,5]. One of the important challenges in this field is the development of the ability for integration of the nanostructured sensing elements into portable electronic devices for point-of-care or point-of-need detection. Most of the existing approaches focus on the CMOS process in terms of complementary metal-oxide-semiconductor technology (CMOS) to address resolution of the resistance-to-digital circuit for the detection of changes in resistance [6,7,8,9,10]. Limited work has been done in electronics circuit board design aimed at an effective coupling to nanostructured chemiresistor arrays towards durable and reliable performances [11,12,13]. A major significance of the chemiresistor array is the enhancement of the sensor performance in selectivity for the detection of multiple analytes. The nanostructured chemiresistor array coupling and device integration require high sensitivity, rapid response, low power-supply, and high durability. Molecularly-linked thin film assemblies of nanoparticles on interdigitated microelectrode platforms feature enhanced sensitivity, selectivity, detection limit, and response time via controlling size, composition, functional group, and spatial properties [14,15,16,17,18,19,20,21], offering the promise for potential applications of the sensors in healthcare and environmental monitoring. However, a key problem for the coupling of the detection electronics with the nanostructured chemiresistor devices is that the measurement current of most commercial instruments is too high for maintaining a good stability of the sensors. Addressing this problem, along with engineering the device portability, multiple channel capability, and wide range measurement, is critical for achieving practical point-of-care or point-of-need biosensor applications. Therefore, the design of an electronic board that can effectively couple the nanostructured chemiresistor arrays is needed for achieving low excitation current and low power consumption so that the potential instability of the sensors can be minimized. In this report, we describe an electronic circuit board in a handheld dimension to couple with the nanostructured chemiresistor array (handheld chemiresistor detector (HCD)) which meets some of the desired performance specifications. This circuit board is integrated with the nanostructured chemiresistor arrays for meeting the optimal performances in device portability, channel multiplexing, measurement range expansion, auto-locking, and sensor stability. The use of a high excitation current in many existing instruments could cause instability in the gold nanoparticle assembly. As such, the use of gold nanoparticle-assembled sensing films for testing the HCD being developed serves as a real-life example to demonstrate the need of the low-current electronic device. The results will be discussed in comparison with the results of some of the commercial benchtop or handheld instruments or meters in interfacing with the nanostructured chemiresistor sensor arrays for detecting volatile organic compounds (VOCs), which is an important research front in developing sensors for point-of-need monitoring of air quality and biosensors for point-of-care breath screening of human health.

## 2. Materials and Methods

### 2.1. Design of Electronic Board to Couple with Chemiresistor Array

The electronic circuit board is designed for the rapid measurement of the resistance (R) changes of the nanostructured chemiresistor arrays. A key design consideration is that the resistance of different sensors in an array may vary greatly, spanning several orders of magnitude. Based on the sensors’ resistance range, the measured resistance is designed to cover a wide range (30 Ω~300 MΩ). The circuit boards are designed in upgradable modules. One basic model of the modules consists of 8 channels, which can be easily modified for expansion to 16, 24, or 32 channels as needed. In order to improve the measurement accuracy of the resistance change ratio (ΔR/R_i_), the measurement range of each channel is subdivided into 16 ranges. The output is the resistance values of the sensors, which is displayed in two ways: (1) by the LCD on the device, and (2) by a computer through the RS232 interface. All electronic components were obtained from Texas Instruments (Dallas, TX, USA). The design and specifications of the electronic circuit board and its performance will be discussed in the Section 3.

### 2.2. Sensor Fabrication

Standard microfabrication methods [21] were used to fabricate the interdigitated microelectrode (IME) devices. A vacuum sputtering deposition system (Nordiko 2000) was used to deposit the thin film electrodes on a glass substrate using the photolithographic method. This method is currently the most widely used method for creating microelectronics on glass or silicon wafer substrates. Gold nanoparticles (2 nm diameter) encapsulated with a monolayer of decanethiolate (DT) were synthesized using a wet-chemical method, and gold nanoparticles of 5-nm diameter were produced by the thermochemical processing method [14]. Different linker and capping molecules were used, including dithiols such as 1,4-butanedithiol (BDT), 1,5-pentaneditjiol (PDT), 1,6-hexanedothiol (HDT), 1,8-octanedithiol (ODT), and 1,9-nonanedithiol (NDT), along with carboxylic acid functionalized thiol, such as 11-mercaptoundecanoic acid (MUA). All chemicals were obtained from Sigma-Aldrich (Milwaukee, WI, USA) unless specified. Hexane (Hx) vapor was generated from hexane solvent (Fisher Scientific, Waltham, MA, USA). The thin films of NDT-linked nanoparticles (NDT-Au_nm_) and MUA-linked nanoparticles (MUA-Au_nm_) were prepared using the “exchanging-cross-linking-precipitation” method [14]. In this method, NDT or MUA exchanges with the alkanethiolates on the gold surface, and then undergoes crosslinking and precipitation via Au-S bonding or hydrogen bonding. The thin films were deposited on the IME devices by the immersion method. The film thickness was controlled by immersion time. The thin films were rinsed thoroughly with hexane before drying under nitrogen gas. The NDT- and MUA-linked nanoparticle thin films on IME are denoted as NDT-Au_nm_/IME, and MUA-Au_nm_/IME, respectively. Other linker-assembled thin films on IME are also labeled similarly.

### 2.3. Sensor Measurement

The low-current, multichannel, handheld device interfaced with nanostructured chemiresistor sensor arrays has been tested under different atmosphere and vapor exposures. The measurements of the sensor resistances were performed using the computer-interfaced HCD and multi-channel multimeter (KMM, Keithley 2700). Both can be interfaced with a computer for data readout. The experiments were carried out at room temperature (22 °C ± 1 °C, and RH ≤ 20%). The vapor was generated by flowing N_2_ gas through a bubbler containing the solvent. The flow rates between 5 and 40 mL/min were used for the mixed vapors, which are combined with N_2_ gas to achieve a flow rate of 100 mL/min. Air and air with controlled relative humidity were also tested as carrier gases for the evaluation of the device performance and the data comparison.

At different vapor concentrations, the measured resistance (R) values were expressed as a differential resistance change, i.e., ΔR/R_i_, where R_i_ is the initial resistance and ΔR is the magnitude of the resistance response. A custom-designed test chamber connected to vapor and N_2_ gas was used to house the IME device. The vapor concentration (ppm (V) was controlled by the mixing ratio of vapor and N_2_ gas (99.99% Airgas, carrier gas) using calibrated mass flow controllers in a custom-built impinger system [14]. In a typical measurement, the test chamber was first purged with N_2_ and then with vapor at a certain concentration for a controlled time period. The multi-channel flow system (Swagelok Modular Platform) features low dead-volume and no cross-contamination for the gas mixing.

## 3. Results and Discussion

### 3.1. Design and Specifications of the Multichannel Electronic Circuit Board

The design of the multichannel electronic circuit board focused on the ability to automatically search for the channel according to the initial resistance value (R_i_) of the sensor in each channel and then lock it in the most appropriate resistance range. This also avoids the difference and time delay of the measured value due to the switching of the resistance value at the boundary of the measurement range. The HCD measures the resistance by measuring the voltage drop upon flowing a constant current through the sensor. The constant current source (6.0 nA to 1.2 mA) for each channel has 16 optional current values, each corresponding to a resistance range. 

Figure 1 shows the block diagram of a basic model of the electronic board, where the dash-line outlined blocks correspond to the different input channels. These channels are configurated as a modular array for the interfacing. Using a multiplexer (MUX), the output voltage signal for the sensor’s resistance measured by each channel is sent to an analog-to-digital converter (ADC), from which the resistance value is calculated by a microcontroller unit (MCU). The design of the electronic board allows for rapid monitoring of each channel of the array sensors. The measured resistance values are transmitted to the readout through the microprocessor interface.

Rational selection of the electronic components is important for an effective adaption to the battery power supply and to meet the low-current and low-power consumption requirements. In the circuit board design, inherent errors of both reference resistance and reference voltage can be eliminated by calibration. However, the temperature drift and the long-term stability of the components may affect the absolute accuracy of the measurement. These selections considered the characteristics of the nanostructured chemiresistor array sensors, the comprehensive measurement accuracy requirement, and the overall device cost. With this circuit design principle, the measurement accuracy of absolute resistance can be further improved by selecting higher-index devices. In principle, the stability and accuracy of the measured values can be further optimized by the layout of the circuit board, the anti-interference capability of the input loop, the influence of the leakage current, the hardware filter components, the software filter algorithms, the sampling rate, etc.

In comparison with commercial benchtop instruments (e.g., Keithley 2700), the main attributes of the handheld device include: (i) Enhanced anti-interference capability and stability under high resistance, and the capability to measure very small resistance change; (ii) Low open circuit voltage with negligible damage to the sensor (note: there was no indication of “breakdown” phenomena for the electronic devices after operating for at least one year); (iii) short boot time (less than 2 min); and (iv) low cost. Table 1 summarizes some of the major specifications in terms of the measurement ranges and errors. Additional technical specifications include: (1) an expandable number of channels; (2) high readout stability (±0.003% (ΔR/R_i_)/min or ±2 Ω/min); (3) a fast sampling rate of each channel (≥4 times/s, for the case of 8 channels working at the same time); (4) automatic searching and locking in terms of the appropriate resistance measurement range; (5) low power consumption (with the backlight off: ≤120 mW); and (6) handheld dimensions and weight (70 × 25 × 200 mm^3^, 350 g).

### 3.2. Performance Characteristic upon Coupling to the Nanostructured Chemiresistor Array

The microelectrode design parameters of the IME devices and the nanostructured thin film assembly on the IME devices are detailed in our previous reports [21]. An 8-sensor array was used in this work. In the sensor array, there are six sensors that were derived from dithiol-linked gold nanoparticles (HDT-Au_2nm_, BDT-Au_2nm_, ODT-Au_5nm_, PDT-Au_5nm_, NDT-Au_2nm_, and HDT-Au_5nm_) and two sensors which were derived from MUA-linked gold nanoparticles (MUA-Au_5nm_, and MUA-Au_2nm_), forming an 8-sensor array [16]. These sensors differ in electrical conductivities due to differences in interparticle distances, particle sizes, dielectric medium properties, and the device design parameters. These chemiresistor sensors behave as pure resistors, as demonstrated in Figure 2A by comparing the I-V curves between pure resistors and the sensors. Linear relationships are evident in the measured current range. By comparing the resistance values obtained from the slopes, there is an insignificant variation for the 5 kΩ resistor (0.22% change). For the 0.1 MΩ cases, there is also an insignificant variation for the 5 kΩ resistor (0.16%). Figure 2B shows a typical set of I–R plots for resistors with different resistance values. Note that the voltage vs current is linear, which follows Ohm’s law. The data can be fitted well with Ohm’s law.

In Figure 3, the results between our electronic circuit board (HCD) and several commercial multimeters (Keithley 2700 (KMM), Agilent Technologies (AT) and BK Precision (BK)) are compared for measuring the resistances of selected resistors and sensors. As shown in Figure 3A, the differences are small when comparing the resistance of physical resistors obtained from the four instruments. However, there is a significant difference for the KMM when measuring the resistances of the sensor from the other three instruments (Figure 3B), reflecting the subtle differences in the excitation current used in the measurement.

### 3.3. Array Responses to VOC and Performance Evaluation

A series of VOCs such as hexane, toluene, ethanol, and water vapors were tested with the electronic board coupled to a nanostructured chemiresistor array with subtle differences in particle sizes (2–5 nm) and molecular linking/capping structures [14,16]. For the performance evaluation of the devices, we focus on the discussion of the results obtained with the chemiresistor array in response to hexane vapor and some pure resistors with comparable resistance values. Figure 4 shows a typical set of the resistance response profiles for an 8-sensor array in response to hexane vapors at several different concentrations. The results are compared between (a) HCD and (b) KMM. It is evident that the response profiles are essentially identical between the two systems, except for the absolute values of the sensor resistances. Note that the same response pattern is assumed by design, since the devices are exposed to the same sequence of vapor concentration increments. While the patterns are the same, the response magnitudes are subtly different depending on the specific sensor element, as shown in Figure 5, which illustrates the subtle differences in the performances of the two instruments. The resistance values from the HCD are higher than those from the KMM. This difference is believed to reflect the electronics board design parameter differences, e.g., the constant current sources. Note that the absolute values of the resistances from KMM is lower than those from HCD, which is consistent with the result in Figure 3B. This result again reflects the subtle difference in the excitation current used by the two different instruments.

Since there are differences in the initial resistance for the different sensing films, the relative resistance change, ΔR/R_i_, was used as a measure of the sensor response signal. Figure 5 shows a typical set of the results for the relative change of the resistance (ΔR/R_i_) of the sensor array in response to hexane vapors of different concentrations between (a) HCD and (b) KMM. Linear correlations with the vapor concentration are essentially identical between the two systems, as reflected by the highly-comparable slope values, i.e., the sensitivities. This result demonstrates comparable sensitivities between the two systems.

The response profiles for different arrays coated with sensing films were also examined for their exposures to several volatile organic vapor analytes. In general, the response characteristic features an increase in ΔR/R_i_ upon exposure to the vapor, followed by a return to baseline upon purging with nitrogen. The response is relatively rapid and reversible. The responses increased linearly with the vapor concentration. The slope serves as a measure of the response sensitivity. The array sensors display linear responses to the concentrations of the VOC. The different sensitivities of the sensor devices reflect the differences in the sensor design parameters, including particle sizes, interparticle linker molecules, and dielectric medium properties.

The noise level of the device was also assessed by examining the noises of the array response data to hexane vapor with different concentrations in comparison with the data obtained using KMM. Based on the results, the detector shows a stability of 0.003%. Note that the initial setting of the minimum resolution of the measurement to 1 Ω could be problematic, since it may be too large for measuring smaller resistances (e.g., for 5 kΩ resistance, 1 Ω/5000 Ω = 0.0002). The jump interval in the digits is related to the inability, which is designed to meet the stability requirement of 0.003%. Higher-performance electronic components, software, and hardware may be needed for handling the noise level exceeding the practical requirement. These measures could further improve the stability index to better than 0.003% (ΔR/R_i_)/min.

In Table 2, the measurement currents and voltages are compared for resistors of different resistances in terms of the design specifications. Both the output current and power consumption for HCD are clearly much smaller than those for KMM when resistances greater than 1 MΩ are measured.

Indeed, experimentally, the baseline drift for the measured resistance of sensors with high resistances was shown to strongly depend on the magnitude of the excitation current and power consumption parameters. As shown in Figure 6 for the measurement of a sensor with a resistance range of 12~13 MΩ, KMM obviously produces a clear baseline drift, whereas HCD showed little drift.

This result is consistent with the fact that the excitation current of KMM is much larger than that of HCD for resistance greater than 1 MΩ. In this example, a sensor with a resistance in the range of 12~13 MΩ was measured. While the measurement took a current of about 12 nA for HCD, a current of 160 nA was observed for KMM, showing 1~2 orders of magnitude difference. As such, the dissipated power (P), which varies exponentially with the square of the current (P = I^2^R), could lead to a heat impact on the sensor. It is known that the resistance of the chemiresistor sensor is sensitive to temperature [22]. A subtle heating impact to the sensor would lead to a decrease in the measured resistance. In this regard, there is a clear advantage for using HCD for the measurement of chemiresistor sensors with high resistance in comparison with many of the commercial instruments.

To further evaluate the performance under ambient conditions, the multichannel device interface sensor array was also tested using air and air with controlled relative humidity (RH) as the carrier gases. Figure 7 shows a representative set of data to compare the response sensitivities of individual sensors in the array in response to hexane vapor in air and air with 30% RH. Note that both room air and pre-prepared air (O2/N2 ratio ~0.26) were used. The pre-mixed air was then humidified with a controlled RH (30%), similar to the room air’s RH.

Figure 7 depicts the differences of the changes in resistance of individual sensors in response to the vapor sorption per unit concentration, which is expected by design, based on the differences in the sensor elements in the array. The result is significant in two aspects. First, the differences of the response sensitivities of different sensors stem from the differences in the interactions between the adsorbed hexane and the sensing films, which is a key element of the design of the sensor array. Second, the variations of the response sensitivities of each sensor between air and air with 30% RH reflect the difference in the sensor for the competitive adsorption of hexane and water vapor in the sensing film, which demonstrates the desired performance of the multichannel device interface sensor array in different sensing environments, as known for the nanostructured chemiresistor sensors interfaced with commercial instrument (KMM) in the detection of mixed VOC and water vapors [17,21].

## 4. Conclusions

In conclusion, we have developed a low-current, multichannel, handheld electronic device integrated with nanostructured chemiresistor sensor arrays. By integrating the circuit board with nanostructured chemiresistor arrays and testing it with VOCs, the handheld device is shown to meet the desired low-current and wide-range performances for different sensors spanning several orders of magnitude in resistance. The combination of the multiplexing, wide-range, auto-locking, and high-stability capabilities is anticipated to find applications in environmental monitoring of VOCs for air quality control and human breath sensing for cancer screening and health monitoring. The current limitation of the HCD is eight channels. Part of our on-going work involves increasing the number of channels to 16 or more. The low excitation current specification also enables low temperature drift and low power consumption so that the potential instability of the chemiresistor sensors can be minimized. This is an important specification, especially for applications of sensors for the point-of-need monitoring of air quality, as well as for biosensors in point-of-care breath screening for human health [23,24,25].

## Figures and Tables

**Figure 1 sensors-22-02781-f001:**
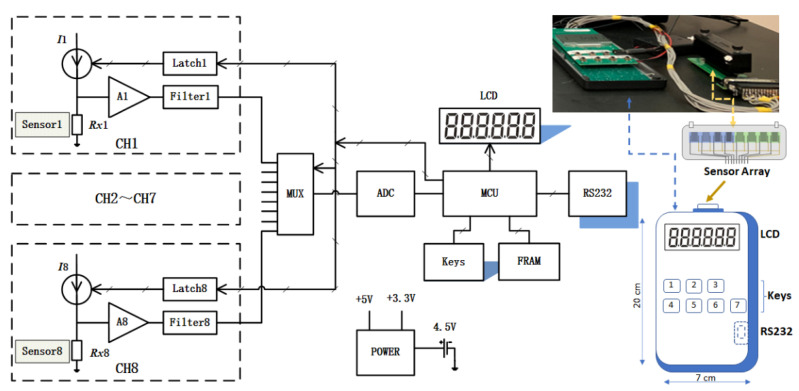
Block diagram for the design of the electronic board for coupling to the nanostructured chemiresistor array. Insets in the right panel: photo of the HCD interface with the sensor array, and the schema showing the layout. Component descriptions—CH1 to CH8: Channel 1 to 8 modules; Sensor1–Sensor8: chemical array sensor 1 to 8 with corresponding resistances from Rx1~Rx2, respectively; I1~I8: Programmable DC constant current source (6 nA~1.2 mA); A1~A8: Operational amplifiers from 1 to 8; Latch1~Latch 8: Range control latches from 1 to 8; Filter1 to Filter8: Filters from 1 to 8; MUX: Multiplexer; ADC: Analog to Digital Converter; MCU: Microcontroller; Keys: Membrane keys with 7 buttons to realize the switch and basic operation functions; LCD: Liquid crystal display, which displays the measured values and the related parameters; Power: DC-DC power module; RS232: Communication interface RS232, which can be interfaced with the Wi-Fi wireless module; FRAM: Ferroelectric random-access memory, which can store the measurement data of the multiple channels at the same time, and depending on the capacity of the selected device, perform continuous sampling up to many days. The stored data can be transmitted to the host computer through RS232 interface. Using FRAM storage mode, the HCD can realize offline field data sampling without an AC power supply. The sensor array is housed in a closed space with flow in/out tubing and is mounted on a circuit board which can be plugged into the HCD connector.

**Figure 2 sensors-22-02781-f002:**
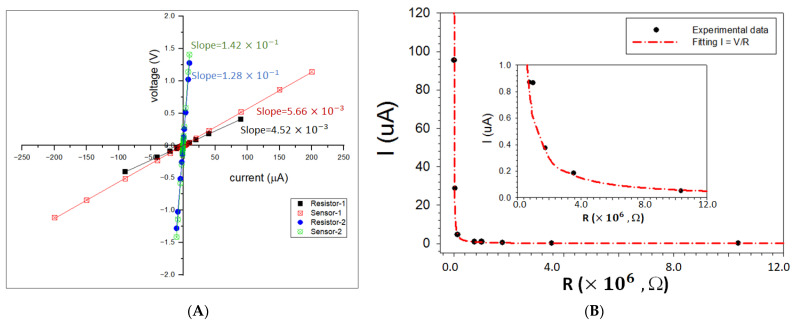
(**A**) Comparisons of I-V curves comparing resistors and sensors obtained using HCD. (Initial values: Resistor-1: R = 4530 Ω; Sensor-1: R = 5700 Ω; Resistor-2: R = 0.12804 MΩ; Sensor-2: R = 0.117 MΩ. Lines: linear regressions with slopes indicated in the plots. (**B**) I–R plots for resistors with different resistance values using HCD: Red dashed line: fitting with Ohm’s law. Inset: a magnified view of the low current region (I < 1 μA).

**Figure 3 sensors-22-02781-f003:**
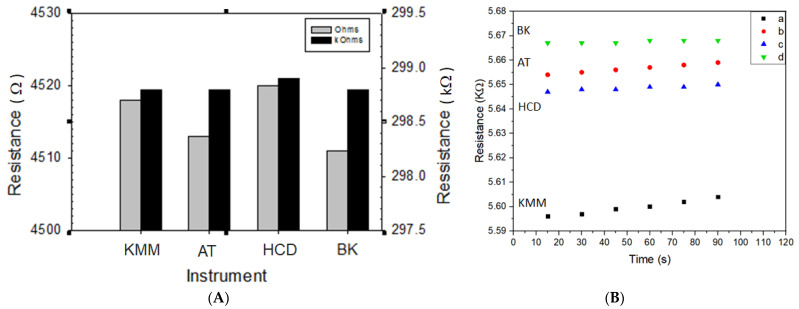
Comparisons of the different instruments in measuring resistance of physical resistors and sensors. (**A**) Comparison of resistance of resistors obtained from four instruments; (**B**) Comparison of resistance of a sensor obtained from four instruments ((a) KMM; (b) AT (Agilent U1252A multimeter); (c) HCD; and (d) BK (B&K Precision 393 Multimeter)).

**Figure 4 sensors-22-02781-f004:**
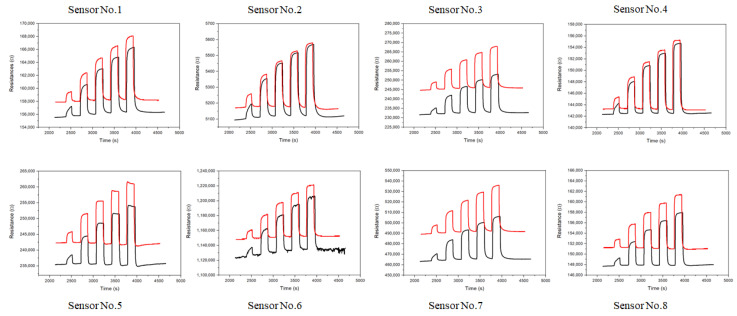
Response profiles of the nanostructured sensor array to hexane vapors of different concentrations with HCD (red curves) and KMM (black curves). The hexane vapor concentration range: 300 ppm~1500 ppm. Sensor array: (**1**) HDT-Au_2nm_, (**2**) BDT-Au_2nm_, (**3**) ODT-Au_5nm_, (**4**) PDT-Au_5nm_, (**5**) NDT-Au_2nm_, (**6**) MUA-Au_2nm_, (**7**) MUA-Au_5nm_, and (**8**) HDT-Au_5nm_.

**Figure 5 sensors-22-02781-f005:**
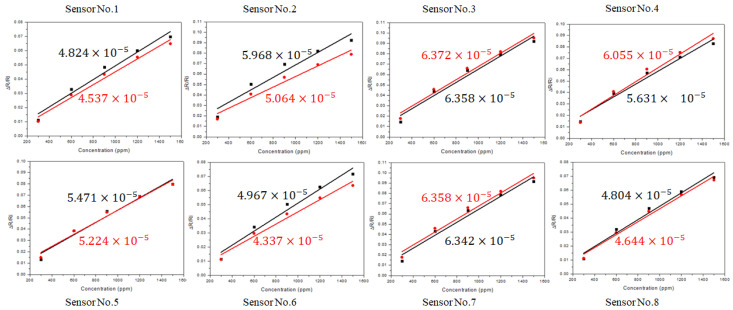
Plots of the responses vs. hexane vapor concentration for the sensor array measured by HCD (red data points) and KMM (black data points). The response sensitivities (ppm^−1^) are indicated by the slopes in the plots.

**Figure 6 sensors-22-02781-f006:**
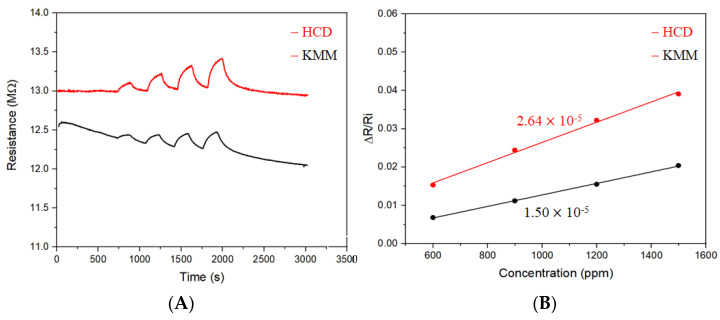
Response profiles (**A**) and sensitivities (**B**) of a chemiresistor sensor in response to hexane vapors of different concentrations obtained using HCD and KMM.

**Figure 7 sensors-22-02781-f007:**
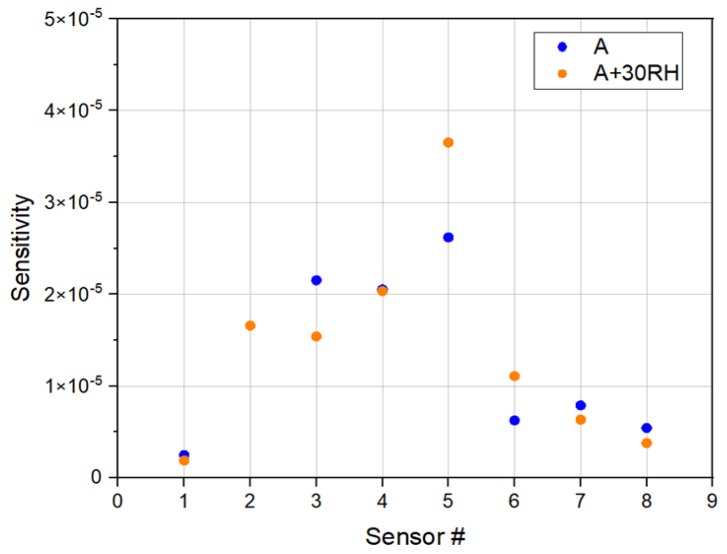
Comparison of the response sensitivities of the individual sensors in the HCD-interfaced sensor array in response to hexane vapor under air and air with a 30% RH sensing environment.

**Table 1 sensors-22-02781-t001:** Specifications of the measurement ranges and errors.

Resistance Range	Resolution	Nominal Current	Max Open Voltage (V)	Accuracy (One Year, TCAL ± 5 °C)
30 Ω~50 kΩ	1 Ω	1.2 mA~29 μA	3.3	±3 Ω
350 kΩ	1 Ω	10 μA~4.7 μA	3.3	±0.05% RD ^1^
4 MΩ	10 Ω	2.4 μA~0.39 μA	3.3	±0.05% RD
30 MΩ	0.2 kΩ	192 nA~55 nA	3.3	±0.2% RD
130 MΩ	1 kΩ	24 nA~12 nA	3.3	±1% RD
300 MΩ	5 kΩ	6 nA	3.3	±2% RD

^1^ Note: RD: Reading Value.

**Table 2 sensors-22-02781-t002:** Comparison of the measurement currents and voltages for resistors of different resistances.

Resistance	KMM		HCD	
	Current	Open Circuit Voltage	Current	Open Circuit Voltage
10 kΩ	100 μA	6.6 V	100 μA	3.3 V
100 kΩ	10 μA	12.8 V	10 μA	3.3 V
1 MΩ	10 μA	12.8 V	1.2 μA	3.3 V
10 MΩ	0.7 μA	7.0 V	0.1 μA	3.3 V
100 MΩ	70 nA	7.0 V	12 nA	3.3 V
300 MΩ	NA	NA	6 nA	3.3 V

Note: The upper limit of resistance for KMM is 100 MΩ.

## Data Availability

Not applicable.

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
