# Peer review of "A Low-Current and Multi-Channel Chemiresistor Array Sensor Device"

_sensors, 2022, doi:10.3390/s22072781_

Round 1

Reviewer 1 Report

The paper presents a design of flow current multichannel and electronically handheld device with key design feature of electronic circuit with low excitation current. The study seems to contribute in the field, however, if the following changes are incorporated it would enhance the impact of the study.

  1. The author in the abstract mentions that, “The device is tested with a chemiresistor array with nanostructured sensing films prepared by molecularly-mediated assemblies of gold nanoparticles for the detection.” To what extent the use of gold nanoparticles contributes to the properties of the device?
  2. The language across the article should be revised and checked for grammar.
  3. What are the practical applications of the device along with economic statistics?
  4. In section 2.2 author mentions the standard microfabrication as the process for making the sensor. Author should clearly explain the technique, purpose of use and comparison with other studies.
  5. Figure-1 quality is very low. Additionally, the captions of the figures throughout the paper should be revised and clearly explained.
  6. In conclusion the limitations and future goal can be mentioned
  7. If the author can mention the significance of chemiresistor array it will be better.
  8. In Figure 5, Sensor 3 and Sensor 7, Sensor 2 and Sensor 6 follow the same pattern. What is the particular reason behind it? Author can support it using excel data or given a proper description.
  9. Figure 7 mentions the sensor vs sensitivity values. What does it actually depict and what effect on the performance of the device it can cause?

Author Response

Reviewer 1:

Comments and Suggestions for Authors

The paper presents a design of flow current multichannel and electronically handheld device with key design feature of electronic circuit with low excitation current. The study seems to contribute in the field, however, if the following changes are incorporated it would enhance the impact of the study.

  1. The author in the abstract mentions that, “The device is tested with a chemiresistor array with nanostructured sensing films prepared by molecularly-mediated assemblies of gold nanoparticles for the detection.” To what extent the use of gold nanoparticles contributes to the properties of the device?

Response:

This is a good question. Most instruments use high excitation current which could cause instability of the gold nanoparticle assembly in the sensing film. In this context, use of gold nanoparticles for the test of the HCD serves as real example to demonstrate the need of the low-current device. In the revised version, we added this statement for the clarification of this point. (page 2)

  1. The language across the article should be revised and checked for grammar.

Response:

Thank you for the suggestion. We have done thorough proof reading of the revised manuscript to ensure the correct use of English and grammar.

  1. What are the practical applications of the device along with economic statistics?

Response:

As discussed in the introduction, the practical applications of the device are the environmental monitoring of VOCs for air quality control and human breath sensing for cancer screening. In the revised version, we further clarified this point. (page 10)

  1. In section 2.2 author mentions the standard microfabrication as the process for making the sensor. Author should clearly explain the technique, purpose of use and comparison with other studies.

Response:

In section 2.2, we did provide some details on the standard microfabrication process for making the sensor, e.g., sputtering deposition system (Nordiko 2000) to deposit the thin film electrodes on glass substrate by photolithographic method.  This method is currently the most widely used method for making microelectronics on glass or silicon wafer substrates. In the revised version, we have added the explanation (page 2). 

  1. Figure-1 quality is very low. Additionally, the captions of the figures throughout the paper should be revised and clearly explained.

Response:

Thank you for the suggestion.  Figure 1 is significantly improved in the revised version. We have also added device diagram (B) in the diagram. We have provided the descriptions of the detailed circuit components in the figure caption. (page 4):

Figure 1. Block diagram for the design of the electronic board for coupling to the nanostructured chemiresistor array. Insets in the right panel: photo of the HCD interface with the sensor array, and the Shechem showing the layout.  Component descriptions: CH1 to CH8: Channel 1 to 8 modules; Sensor1-Sensor8: chemical array sensor 1 to 8 with corresponding resistances from Rx1~Rx2, respectively; I1~I8: Programmable DC constant current source (6 nA~1.2 mA);  A1~A8: Operational amplifiers from 1 to 8; Latch1~Latch 8: Range control latches from 1 to 8;  Filter1 to Filter8: Filters from 1 to 8;  MUX: Multiplexer;  ADC: Analog to Digital Converter; MCU: Microcontroller; Keys: Membrane keys with 7 buttons to realize the switch and basic operation functions;  LCD: Liquid crystal display, which displays the measured values and the related parameters;  Power: DC-DC power module;  RS232: Communication interface RS232, which can be interfaced WI-FI wireless module.  FRAM: Ferroelectric random-access memory, which can store the measurement data of the multiple channels at the same time, and depending on the capacity of the selected device, perform continuous sampling up to many days. The stored data can be transmitted to the host computer through RS232 interface. Using FRAM storage mode, the HCD can realize offline field data sampling without AC power supply.

  1. In conclusion the limitations and future goal can be mentioned

Response:

Thank you for the suggestion.  The current limitation of the HCD is 8 channels.  Part of our future work involves increase of the channels to 16 and more. We have mentioned the limitations and future goal in the conclusion section of the revised manuscript (page 10).

  1. If the author can mention the significance of chemiresistor array it will be better.

Response:

Thank you for the suggestion. A major significance of chemiresistor array is the enhancement of the sensor performance in selectivity for detection of multiple analytes.  The significance of chemiresistor array is discussed in the revised version. (page 1-2).

  1. In Figure 5, Sensor 3 and Sensor 7, Sensor 2 and Sensor 6 follow the same pattern. What is the particular reason behind it? Author can support it using excel data or given a proper description.

Response:

Yes, following the same pattern is supposed to be so by design since they are exposed to the same sequence of vapor concentration increments. While the patterns are the same, their response magnitudes are subtly different depending on the specific sensor element, as shown in Figure 5, which illustrates the subtle differences in the performances of the two instruments. We have added the description in the revised version. (page 7)

  1. Figure 7 mentions the sensor vs sensitivity values. What does it actually depict and what effect on the performance of the device it can cause?

Response:

The sensitivity shown in  Figure 7 is explained in the description following the Figure 7. For a further clarification, it depicts the differences of the changes in resistance of individual sensors in response to the vapor sorption per unit concentration, which is expected by design based on the differences in the sensor elements in the array. We have added the description in the revised version. (page 10)

Reviewer 2 Report

This article is possibly more suited to IEEE sensors, but interesting enough to warrant publication in Sensors.  The article presents an approach to handheld electronics for use with nanoparticle based chemiresistors and demonstrates application, using VOCs as a testbed.  The article is interesting and important for the field to move towards practical devices.  There are a couple of areas that can be better explained, and a couple of typos, but the main critique is that the sensor flow cell arrangement is not shown.  The article is about hardware, but the schematic drawing in figure 1 doesn’t give a description of how the 8 sensors are implemented.  There is a picture, but the image is very small, difficult to view, and not explained.  At the very least, the authors should add supplemental information on how the sensor hardware was implemented.  Are the sensors mounted on TO packages? How are the sensors and electronic hardware interconnected?

Areas needing better explanation:

  1. What is the output of the devices? A digital display of resistance?
  2. How is the Keithley instrument used? It appears to be used as a reference to compare with their handheld unit, but the statements on line 97 & 98 seem to imply that the KMM was used as the computer interface. This is confusing.
  3. Lines 99-101 describe vapor mixing, but then lines 110 – 112 in the next paragraph say a different thing about how vapor mixing occurs. These two descriptions should be combined into one clear description.
  4. Air and Air with 30% RH are used, but it isn’t clear what air is. Is it a synthetic air with 0% humidity, or room air with unknown RH? Is the RH > or < 30%? 
  5. What is the readout – line 133? Is this a digital display of resistance?
  6. Figure 1 is partially explained, but what is Keyboard, Fram, Interface, and the last symbol on the right (RS-232?)? Is the handheld unit complete, or does it require a computer connection for data processing and display?
  7. Figure 2B says that it follows Ohms law, but the curve is not linear. Presumably, the voltages vary, but this isn’t explained.
  8. Figure 7 compares sensitivity data for Air and 30% RH, but the values have little meaning. It would be beneficial to add the N2 data to that plot as well so that the effect of air is clear. Also, do the pulses in air look like Figure 4?

Typos:

Line 109: custom-designed.

Line 168: size sensors?

Line 179: 5 KOhm doesn’t match line 178 0.1 MOhm

Figure 3 has an extra line below (B).

Line 250 doesn’t make sense

Line 271 has an extra space

Author Response

Reviewer 2:

Comments and Suggestions for Authors

This article is possibly more suited to IEEE sensors, but interesting enough to warrant publication in Sensors.  The article presents an approach to handheld electronics for use with nanoparticle based chemiresistors and demonstrates application, using VOCs as a testbed.  The article is interesting and important for the field to move towards practical devices.  There are a couple of areas that can be better explained, and a couple of typos, but the main critique is that the sensor flow cell arrangement is not shown.  The article is about hardware, but the schematic drawing in figure 1 doesn’t give a description of how the 8 sensors are implemented.  There is a picture, but the image is very small, difficult to view, and not explained.  At the very least, the authors should add supplemental information on how the sensor hardware was implemented.  Are the sensors mounted on TO packages? How are the sensors and electronic hardware interconnected?

Response:

Thank you for the questions and suggestion. The sensor flow cell and hardware interfacing and implementation are added into Figure 1. The sensor array is housed in a closed space with flow in/out tubing and is amounted on a circuit board which can be plugged onto the HCD connector. These descriptions are included in Figure 1’s caption of the revised manuscript (page 4).

Areas needing better explanation:

  1. What is the output of the devices? A digital display of resistance?

Response:

Thank you for the questions. The output is the resistance values of all sensors, which is displayed in two ways: (1) by the LCD on the device, and (2) by a computer through the RS232 interface. We have added the description in the revised version. (page 2)

  1. How is the Keithley instrument used? It appears to be used as a reference to compare with their handheld unit, but the statements on line 97 & 98 seem to imply that the KMM was used as the computer interface. This is confusing.

Response:

Keithley instrument was used the same as the HCD so we could compare the results. Both can be interfaced with the computer for data readout. We have added the description in the revised version for clarification. (page 3)

  1. Lines 99-101 describe vapor mixing, but then lines 110 – 112 in the next paragraph say a different thing about how vapor mixing occurs. These two descriptions should be combined into one clear description.

Response:

Thanks for pointing the redundancy out.  This is fixed in the revised version. (page 3)

  1. Air and Air with 30% RH are used, but it isn’t clear what air is. Is it a synthetic air with 0% humidity, or room air with unknown RH? Is the RH > or < 30%? 

Response:

We used both room air and pre-prepared air (O2/N2 ratio ~ 0.26) by Air Gas company. The pre-mixed air was then humidified with a controlled RH similar to the room air’s RH (~30%).  We have added the description in the revised version for clarification. (page 9)

  1. What is the readout – line 133? Is this a digital display of resistance?

Response:

The device is designed to feature both digital and computer- readout.

  1. Figure 1 is partially explained, but what is Keyboard, Fram, Interface, and the last symbol on the right (RS-232?)? Is the handheld unit complete, or does it require a computer connection for data processing and display?

Response:

Thank you for the questions.   RS232: Communication interface RS232, which can be interfaced WI-FI wireless module.  FRAM: Ferroelectric random-access memory, which can store the measurement data of the multiple channels at the same time, and depending on the capacity of the selected device, perform continuous sampling up to many days. The stored data can be transmitted to the host computer through RS232 interface. Using FRAM storage mode, the HCD can realize offline field data sampling without AC power supply.   The HCD is complete unit, and it can be interfaced with a computer through the RS232. We have included the information in the caption of Figure 1 of the revised version (page 4).

  1. Figure 2B says that it follows Ohms law, but the curve is not linear. Presumably, the voltages vary, but this isn’t explained.

Response:

Yes, Figure 2B shows I vs. R plot. But the voltage vs current is linear, as shown in Figure 2A. It follows Ohms law. We have included the description in the revised version for clarification (page 5).

  1. Figure 7 compares sensitivity data for Air and 30% RH, but the values have little meaning. It would be beneficial to add the N2 data to that plot as well so that the effect of air is clear. Also, do the pulses in air look like Figure 4?

Response:

Figure 7 is intended to compare the sensitivity data obtained under two different testing conditions in response to hexane, air and air with 30% RH. The meaning is described in the manuscript. We have included the description in the revised version for clarification (page 9-10). We have done data collection using N2 as background gas. The result was essentially the same. Pulses in air causes quick and small disturbances, which do not affect the VOC response profile.  Since the comparison is intended for practical application, we presented the data using air as the background gas. 

Typos:

Line 109: custom-designed.

Line 168: size sensors?

Line 179: 5 KOhm doesn’t match line 178 0.1 MOhm

Figure 3 has an extra line below (B).

Line 250 doesn’t make sense

Line 271 has an extra space

Response:

Thank you for pointing out these typos. They are corrected in the revised version.

Reviewer 3 Report

The paper entitled "A Low-current and Multi-channel Chemiresistor Array Sensor Device" deal with the realization of a new design for a device for the investigation of nanostructured chemiresistor sensor array. The authors faced several problems that affect the classical devices and created a device with a very high promising aspectatives. 
Nevertherless the paper in my eyes needs a minor revision. Mainly related to some explaination for a clearer understanding of readers. 

In particular: 
First time that the acronim VOC is used, exaplaine it fully. 

Results section, line 176-180. (I believe that values of variations should be checked and corrected.)
Fig. 2 typos in caption. I would suggest here to improve the description about how the variations are calculated. 

Fig. 3b, insert in the legend the devices used, not only in the caption.
Line 205 is specified that several VOC where investigated but only hexane vapor is reported. 

Fig. 4, I suggest to include the concentrations in the plots. 

Line 257 is declaring that current and power consumption for HCD are clearly much smaller than for KMM when 1MOhm are measured. In fact, in the table is reported a current of 0.07 uA vs 12 nA for 100MOhm, but for a better reading I would suggest to put the 2 values in the same scale, 70nA and 12 nA, and specify in the text that the difference is a factor 6. 

Author Response

Reviewer 3:

Comments and Suggestions for Authors

The paper entitled "A Low-current and Multi-channel Chemiresistor Array Sensor Device" deal with the realization of a new design for a device for the investigation of nanostructured chemiresistor sensor array. The authors faced several problems that affect the classical devices and created a device with a very high promising aspectatives. 
Nevertherless the paper in my eyes needs a minor revision. Mainly related to some explaination for a clearer understanding of readers. 

In particular: 
First time that the acronim VOC is used, exaplaine it fully. 

Response:

It is explained in the introduction section manuscript: volatile organic compounds (VOCs)

Results section, line 176-180. (I believe that values of variations should be checked and corrected.)
Response:

Thank you for pointing it out. There are some mislabeling and redundancy, which are fixed in the revised version (page 5).

Fig. 2 typos in caption. I would suggest here to improve the description about how the variations are calculated. 

Response:

Thank you for pointing it out. We have corrected the typos, and added description for better clarification (page 6).

Fig. 3b, insert in the legend the devices used, not only in the caption.
Response:

Thank you for the suggestion. We have inserted the legends in the updated Fig. 3B.

Line 205 is specified that several VOC where investigated but only hexane vapor is reported. 

Response:

Yes, we tested several VOCs. But his paper we focused on the comparison of the electronic devices’ performance comparison, and thus used only the data for hexane for the comparison.

Fig. 4, I suggest to include the concentrations in the plots. 

Response:

While this is good suggestion, it would lead too many redundancies in the plots. We decided to include concentrations in the figure caption.

Line 257 is declaring that current and power consumption for HCD are clearly much smaller than for KMM when 1MOhm are measured. In fact, in the table is reported a current of 0.07 uA vs 12 nA for 100MOhm, but for a better reading I would suggest to put the 2 values in the same scale, 70nA and 12 nA, and specify in the text that the difference is a factor 6. 

Response:

Thank you for the suggestion, which is well taken in the revision:

Round 2

Reviewer 1 Report

The manuscript's quality has been significantly improved. I recommend its acceptance.